# Advancements in Drug Repurposing: Examples in Psychiatric Medications

**DOI:** 10.3390/ijms241311000

**Published:** 2023-07-01

**Authors:** Ryo Okuyama

**Affiliations:** College of International Management, Ritsumeikan Asia Pacific University, Beppu 874-8577, Japan; ryooku@apu.ac.jp

**Keywords:** psychiatric disease, drug repurposing, drug development

## Abstract

Because there are a limited number of animal models for psychiatric diseases that can be extrapolated to humans, drug repurposing has been actively pursued. This study was aimed at uncovering recent trends in drug repurposing approaches and new technologies that can predict efficacy on humans based on animal models used in psychiatric drug development. Psychiatric drugs that were approved by the FDA between 2002 and 2022 were listed, and the method of how the drug repurposing has been applied was analyzed. Drug repurposing has been increasingly applied to recently approved psychiatric drugs. The development concepts of psychiatric drugs that have been developed through drug repurposing over the past 20 years were found to be divided into six categories: new application exploration, reduction of side effects, improvement of symptom control, improvement of medication compliance, enhancement of drug efficacy, and reduction of drug–drug interactions. All repurposed drugs approved before 2016 used either prodrugs or active metabolites, while all drugs approved in 2021 and beyond used fixed-dose combinations with sophisticated ideas. SmartCube^®^, which uses artificial intelligence to predict human drug efficacy from animal phenotypes, was developed and produced novel drugs that show clinical efficacy. Well-designed drug repurposing approaches and new technologies for predicting human drug efficacy based off of animal models would contribute to novel psychiatric drug development.

## 1. Introduction

A drug candidate that was discovered in the research phase has an approximately 10% likelihood of gaining regulatory approval following clinical trials [1]. This is partly due to the difficulty in accurately predicting the efficacy and safety of drugs in humans from non-clinical tests. Approximately 80% of drug candidates fail in the clinical development stage due to inadequacies regarding the efficacy, side effects, and pharmacokinetics [2], making it extremely important to accurately estimate the efficacy and safety of new drug candidates for humans starting from the non-clinical stage. In addition, the cost of developing a single new drug until approval is increasing annually; currently, the cost exceeds USD 1 billion [3]. Therefore, increasing the success rate of new drug development in clinical trials by improving the extrapolation from the non-clinical stage to humans is a critical issue for pharmaceutical companies for their sustainable growth and survival.

Extrapolation from non-clinical to human settings is particularly important in disease areas where there is a lack of animal models that accurately reflect humans. Psychiatric disorders are complex diseases in which multiple mechanisms are involved in disease pathogenesis and drug efficacy. Furthermore, there are significant differences in brain structure and function between animals and humans, which highlight the weaknesses of animal models compared with other disease areas [4]. Therefore, drug repurposing, which involves utilizing already approved drugs or clinically developed compounds, has been widely practiced in psychiatric drug development [5,6]. An approved drug has proven safety and pharmacokinetics in humans, lowering the risk of clinical development failure for a new disease, especially in therapeutic areas where disease pathophysiology and the drug’s mechanism of action are complex.

Drug repurposing is the drug development approach of discovering new indications and/or new treatment options for approved drugs or development compounds that have failed in clinical trials. Their new indications have been discovered serendipitously in many cases, and no systematic approach has been established in the process [7]. There are three types of drug repurposing: repositioning, reformulation, and combination [6]. Repositioning refers to finding a different application for a drug that was originally developed and approved for a different indication. One notable example is sildenafil. Initially developed as a medication for hypertension, it was later discovered through clinical use to be effective for erectile dysfunction, leading to its approval for that application as well [8]. Thalidomide is another famous example of drug repositioning. Thalidomide was developed for the treatment of sedation in the 1950s and was withdrawn because of its teratogenicity. Subsequent research revealed the diverse effects of thalidomide, leading to its repositioning for multiple myeloma and other types of tumors [9]. Reformulation is a method of improving the drug profile by changing the route of administration or by creating prodrugs. Amiodarone, an antiarrhythmic drug, is an example of improving drug profiles through a formulation change. Patients who do not achieve sufficient effects with oral amiodarone were believed to have insufficient drug concentrations distributed in the myocardium, which led to the development of intravenous (IV) amiodarone to address this issue [10]. It has become common to administer IV amiodarone to patients who continue to experience recurrent arrhythmias despite taking oral amiodarone in order to improve their symptoms [10]. Minoxidil is an example where both drug repositioning and reformulation have been applied. Minoxidil in an oral formulation has been used to treat severe hypertension. While being used clinically, hair growth has been reported as a side effect of Minoxidil. This finding led to the subsequent development of a topical Minoxidil formulation for the treatment of androgenetic alopecia [11]. Combination refers to the method of improving the drug profile by creating a combination of two or more drugs. There are some examples present in recent cardiovascular drugs; examples include the combination of acetazolamide and loop diuretics for acute decompensated heart failure, the combination of statins and ezetimibe for atherosclerosis, and the combination of angiotensin receptor blockers and β-blockers for Marfan syndrome [12].

The purpose of this study was to investigate the recent trends in drug repurposing in psychiatric diseases and suggest directions for the development of therapeutic drugs. How has the importance of drug repurposing changed in psychiatric diseases? How has the methodology of drug repurposing evolved? Are there any new initiatives to enhance extrapolation to humans in psychiatric drug discovery and development? To answer these questions, this study examined psychiatric drugs that were approved by the US Food and Drug Administration (FDA, Silver Spring, MD, USA) over the past 20 years and explored the trends in drug repurposing. Additionally, the study reports on the SmartCube^®^ framework, a novel system used for predicting human responses from animal behaviors in psychiatric drug discovery, and discusses its significance.

## 2. Results

### 2.1. Use of Human Phenotype Information of Psychiatric Drugs Approved by the FDA in the Past 20 Years

To investigate the chronological trend of drug repurposing for approved psychiatric drugs, the FDA-approved drugs for psychiatric diseases in the past 20 years were listed; drugs whose active ingredient had been used previously in humans were selected. Of the new drugs approved by the FDA between 2002 and 2022, 30 were approved for use in psychiatric diseases. Table 1 lists the date of approval, brand name, active ingredient, applicable disease, and prior use of each drug in humans (and if so, the details of each drug). Ten of the drugs had been previously used for different purposes (Auvelity, Lybalvi, Qelbree, Azstarys. Diacomit, Epidioloex, Lucemyra, Aristada, Pristiq, and Invega). Among the 10 drugs whose active ingredient had been used previously in humans, only 3 drugs were approved before 2017, and 7 drugs were approved in the past 5 years (2018–2022).

### 2.2. Development Concepts and Compound Approaches

The development concepts of psychiatric drugs that were developed utilizing compounds that were previously administered to humans and approved by the FDA from 2002 to 2022 were analyzed. As a result, the development concepts could be classified into six categories (Table 2).

The first category is the concept of developing new indications for existing compounds. This involves seeking alternative applications for compounds that have already been approved or clinically developed for other diseases. Among the drugs identified in this study, Qelbree, Lucemyra, and Diacomit fall under this category. Additionally, Epidiolex is an example where the development for epilepsy was conducted based on clinical observations of cannabidiol improving severe seizures in humans [13]; it was also considered to fall under the concept of developing new indications for existing compounds.

The second category is the concept of reducing the side effects of existing drugs. In the case of Invega, the development concept was to reduce the side effects by using a sustained-release formulation of the active metabolite while maintaining the therapeutic effects of the existing drug [14]. Lybalvi is a drug aimed at reducing the weight gain side effect associated with olanzapine, which was already prescribed for schizophrenia and bipolar disorder, by combining it with samidorphan, an opioid antagonist that was clinically developed for humans before [15]. Additionally, although not yet approved, the example of KarXT falls into this category as well. KarXT is a fixed-dose combination of xanomeline, a selective M1 and M4 receptor agonist, and trospium, a peripheral muscarinic receptor antagonist. Xanomeline was clinically developed for schizophrenia and showed good efficacy, but its development failed due to gastrointestinal side effects resulting from peripheral muscarinic receptor stimulation [16,17]. Trospium is an approved peripheral muscarinic receptor antagonist [18], and the concept of developing KarXT was to reduce the peripheral side effects of xanomeline by combining it with trospium [19]. KarXT showed promising results in phase 3 (Ph3) clinical trials [20].

The third category is the concept of improving symptom control by enhancing pharmacokinetics. Azstarys is a drug that combines the active moiety with its prodrug, creating a pharmacokinetic profile that achieves a rapid onset and sustained effect, contributing to the improvement of symptom control [21].

The fourth category is the concept of reducing the dosing frequency and improving medication compliance through prodrug formulation. Aristada enables multiple dosing interval options by utilizing a prodrug with sustained release [22]. Pristiq, a prodrug of the approved antidepressant venlafaxine, allows for once-daily dosing, thus improving medication compliance over venlafaxine [23].

The fifth category is the concept of enhancing therapeutic efficacy through combination therapy. Auvelity is a combination of dextromethorphan and bupropion. Dextromethorphan has long been clinically prescribed as an antitussive, but it is readily metabolized by CYP2D6 [24,25]. Dextromethorphan is considered a N-methyl-D-aspartate antagonist, and its antidepressant effect was expected based on its mechanism of action [26]. However, its short half-life posed a challenge. To address this, the concept behind Auvelity was to combine dextromethorphan with bupropion, which is already prescribed as an antidepressant in clinical practice and is also known as a CYP2D6 inhibitor [27,28]. This fixed-dose combination aims to enhance the sustained effect by prolonging the action of dextromethorphan and by exerting a potent antidepressant effect through the actions of both compounds [29].

The sixth category is the concept of reducing drug–drug interactions. Pristiq achieves this by attenuating drug interactions compared with its active form, venlafaxine, through prodrug modification [30].

Among the compounds approved from 2006 to 2015 that realize the aforementioned development concepts, prodrug modification of approved drugs and utilization of active metabolites were employed (Table 2). On the other hand, for drugs approved in 2021 or later and for KarXT, which is in phase 3, a fixed-dose combination approach was taken (Table 2). Auvelity utilized a combination of compounds that contribute to both therapeutic efficacy and the inhibition of compound metabolism. Lybalvi and KarXT employed a well-designed approach by creating combinations with compounds that possess a different mechanism of action capable of suppressing the side effects of the active compound.

### 2.3. A New Methodology for Predicting Human Therapeutic Efficacy from Animal Phenotypes

In addition to the drug repurposing efforts, a new approach to psychiatric drug discovery has emerged in recent years. This approach involves predicting the therapeutic efficacy that a compound will exhibit in humans based on the phenotype changes observed when the compound is administered to animals. This methodology, known as SmartCube^®^, is valuable in the challenging field of psychiatric drug development, where predicting human therapeutic efficacy from animal phenotypes is particularly difficult [31]. The following provides a detailed description of this methodology.

#### SmartCube^®^

SmartCube^®^ is an animal assay system developed by PsychoGenics to discover drugs for psychiatric diseases [31]. In psychiatric diseases, multiple neural circuits are involved in the pathophysiology and drug action, and drug discovery methods based on animal phenotypes rather than molecular mechanisms have been effective, especially for first-in-class drugs [32]. SmartCube^®^ is a novel drug discovery tool that uses a number of drug compounds that have been marketed or reported to be effective against specific neurological diseases as reference compounds in mice and analyzes the phenotypic features of the mice using machine learning [31]. By comparing the phenotypic features of mice treated with the reference compounds and a new compound, the system predicts the effects of the new compound on the human nervous system [33,34]. Approximately 500,000 data points of mouse phenotype information are collected in an assay, and these data are expressed as 2000 features related to locomotion, trajectory complexity, body posture, and shape [33,34]. This enables us to predict the efficacy of new compounds in human psychiatric diseases, overcoming species differences between animals and humans. In addition, because the pharmacological effects of drugs for psychiatric diseases vary depending on the administered dose, an analysis of each dose allows for highly accurate predictions [34]. PsychoGenics used SmartCube^®^ to evaluate the efficacy of eltoprazine (a partial 5HT1A/1B agonist; this mechanism was not known to be effective for attention deficit hyperactivity disorder (ADHD)) in ADHD and proved its efficacy in humans by conducting a proof-of-concept study in patients with ADHD [34].

SmartCube^®^ is being used in drug discovery for psychiatric diseases and is contributing to the development of psychiatric drugs with novel mechanisms that show efficacy for humans. Sumitomo Dainippon Pharma reported that SEP-363856, a novel treatment for schizophrenia that was identified by its subsidiary Sunovion Pharmaceuticals Inc. using SmartCube^®^, displayed favorable efficacy in a phase 2 study and received the breakthrough therapy designation from the FDA [35]. Previous drugs for schizophrenia acted on dopamine D2 receptors; however, SEP-363856 is a first-in-class therapy that is expected to be effective for patients with schizophrenia who are inadequately treated with existing drugs [36]. Otsuka Pharmaceutical announced that it has collaborated with Sumitomo Dainippon Pharma for four new drug candidates under development for psychiatric diseases in 2021 [37]. Three of the compounds, including SEP-363856, were identified using the SmartCube^®^ system [37]. At the publication level, a novel α4β2-nAChR ligand, a glycogen synthase kinase-3 inhibitor, and a histone deacetylase 6 inhibitor have been identified using SmartCube^®^ [38,39,40], highlighting its potential for the discovery of new psychiatric drugs.

## 3. Discussion

Psychiatric drug development suffers from low translatability from animals to humans, and drug repurposing has been actively pursued to solve this problem. In this study, 10 of the 30 psychiatric drugs that were approved by the FDA between 2002 and 2022 were drugs whose active ingredients were previously used in humans. The percentage of drug repurposing has been particularly high in recent years, with 2 out of the 15 drugs approved between 2002 and 2014 and 8 out of the 15 drugs approved between 2015 and 2022 utilizing drug repurposing, representing more than half of the total. This indicates that drug repurposing methods are becoming increasingly important in the development of drugs for psychiatric diseases.

This study revealed that drug repurposing in psychiatric diseases is not limited to the conventional concept of drug “repositioning”, which is the search for new applications of existing drugs, but is aimed at a variety of development concepts. Drug development was conducted based on the concepts of reducing the side effects of existing drugs, improving symptom control, improving drug compliance, enhancing drug efficacy, and reducing drug–drug interactions.

Another unique finding in these study data is that the commonly employed approaches included the application of existing compounds to other diseases, prodrug formation, and the utilization of active metabolites in the case of psychiatric drugs approved before 2019; for drugs approved in 2021 and onwards, fixed-dose combinations were predominantly utilized. Conventional drug repositioning approaches are still being used in recent years, and the fixed-dose combination strategy has been employed for a long time. However, there is a noticeable trend of changing approaches in drug repurposing for psychiatric disease treatments. These approaches were: combining compounds that can provide both efficacy and metabolic inhibition to existing drugs; reducing the side effects of existing drugs by combining compounds with a different mechanism of action; and creating drugs with optimized pharmacokinetics by combining the active moiety and prodrug. All of these approaches are unique compared with simple methods such as drug repositioning and prodrug creation, indicating that drug repurposing methods are becoming more sophisticated. Because there is a limit to the number of compounds that have been administered to humans, it is thought that simple application expansion and prodrug creation approaches alone have exhausted ideas, and more ingenious methodologies have been sought. The future development of drugs for psychiatric diseases will require even more flexible ideas that are not bound by conventional approaches.

In addition to the utilization of compounds administered to humans, the emergence of the Smartcube^®^ system, which uses artificial intelligence to predict the drug efficacy in humans based on a phenotypic analysis of animals, will provide new opportunities for the development of psychiatric drugs [31]. The fact that SEP-363856, a novel compound discovered using SmartCube^®^, achieved a proof-of-concept for schizophrenia with a first-in-class mechanism is evidence of the usefulness of this assay system [35,36]. Smartcube^®^ can predict human drug efficacy profiles with high accuracy, even for compounds in which no human dosing information is available. Especially in the case of novel mechanisms, it is difficult to predict drug efficacy in humans. Smartcube^®^ will make a significant contribution to the effective translation of drugs to humans in psychiatric diseases, particularly where there are few animal models that can be extrapolated to humans. However, whether the feasibility of this method will be fully established is still unclear. SEP-363856 is currently the only compound that achieved a clinical proof-of-concept, and other compounds that were reported in the literature are still in the preclinical stage [38,39,40]. More evidence is needed to validate the effectiveness of this method.

One limitation of this study is that the analysis did not include off-label use of approved drugs. Off-label use of the drugs have been relatively common in psychiatric disease treatments but was not considered in this study. Another limitation is that it does not include some cases that could be considered drug repurposing. For example, esketamine, approved in 2019 for treatment-resistant depression, is an enantiomer of ketamine; ketamine has been used as an anesthetic. However, esketamine itself as a compound has not been previously clinically utilized and was not considered as drug repurposing in this study. Another limitation is country differences. This study only focused on FDA-approved drugs; therefore, the approved indications of the drugs in other countries were not included in the analysis.

## 4. Materials and Methods

New drugs approved between 2002 and 2022 were searched for on the FDA homepage, including the New Drugs at FDA website (https://www.fda.gov/drugs/development-approval-process-drugs/new-drugs-fda-cders-new-molecular-entities-and-new-therapeutic-biological-products, accessed on 12 January 2023). This search targeted drugs for which the approved applicable diseases were psychiatric disorders. The active ingredient of each listed drug was searched for in the Orange Book (https://www.accessdata.fda.gov/scripts/cder/ob/index.cfm, accessed on 12 January 2023) to determine whether it had been previously approved for another indication. Active ingredients were also searched for using articles and the Internet to determine if they had been in clinical development for other purposes. The development concepts and compound approaches for these identified drugs were investigated through a literature search using the Web of Science database. The information on SmartCube^®^ was obtained by searching Web of Science using the keyword “SmartCube” to collect relevant literature. In addition, the term was searched for on the Internet to collect relevant information, and necessary information was added to describe the case.

## 5. Conclusions

In the field of psychiatric diseases, where extrapolation from animals to humans is difficult, new drug development using drug repurposing has been increasing in recent years. Drug repurposing aims to not only explore new applications but also reduce side effects, improve symptom control, improve drug compliance, enhance drug efficacy, and reduce drug–drug interactions. Until 2018, the methods employed were mainly focused on formulating prodrugs and on exploring different applications of the compounds; however, more recently, the focus has been on fixed-dose combinations. Fixed-dose combinations have taken ingenious approaches such as by combining the active compound with other existing drugs that have mechanisms to reduce degradation or side effects or by combining the active moiety with a prodrug. As a way to improve the accuracy of predicting human drug efficacy, SmartCube^®^, which uses artificial intelligence to predict human drug efficacy from animal phenotypes, has been developed and has successfully created drugs with novel mechanisms, which have obtained clinical proof-of-concept status. It is expected that sophisticated drug repurposing approaches and a new assay system for predicting human drug efficacy from non-clinical data will accelerate the discovery and development of new psychiatric drugs.

## Figures and Tables

**Table 1 ijms-24-11000-t001:** FDA-approved drugs for psychiatric diseases between 2002 and 2022 and their previous administration in humans for different indications.

Approved Date	Brand Name	Active Ingredient	Indication	Active Ingredient Used on Humans before?	Details
20 August 2022	Auvelity	Dextromethorphan and bupropion	Major depressive disorder	Yes	Dextromethorphan has been used as an antitussive. Bupropion has been used as an antidepressant.
28 May 2021	Lybalvi	Olanzapine and samidorphan	Schizophrenia and certain aspects of bipolar I disorder	Yes	Olanzapine has been used for schizophrenia and bipolar disorder. Samidorphan was clinically developed and discontinued before.
2 April 2021	Qelbree	Viloxazine	Attention deficit hyperactivity disorder	Yes	Used for depression for thirty years, then was discontinued
2 March 2021	Azstarys	Serdexmethylphenidate and dexmethylphenidate	Attention deficit hyperactivity disorder	Yes	Dexmethylphenidate has been used for ADHD starting from 2002.
20 December 2019	Caplyta	Lumateperone tosylate	Schizophrenia	No	
21 November 2019	Xcopri	Cenobamate	Partial onset seizures	No	
19 March 2019	Zulresso	Brexanolone	Postpartum depression	No	
5 March 2019	Spravato	Esketamine	Treatment-resistant depression	No	
20 August 2018	Diacomit	Stiripentol	Seizures associated with Dravet syndrome	Yes	Developed for adults with focal seizures and failed in phase 3 before
25 June 2018	Epidioloex	Cannabidiol	Rare, severe forms of epilepsy	Yes	Main component of cannabis plant that is taken by humans
16 May 2018	Lucemyra	Lofexidine hydrochloride	Non-opioid treatment for management of opioid withdrawal symptoms	Yes	Historically used to treat high blood pressure
18 February 2016	Briviact	Brivaracetam	Partial onset seizures	No	
5 October 2015	Aristada	Aripiprazole lauroxil	Schizophrenia	Yes	N-acyloxymethyl prodrug of aripiprazole, a long-acting injectable atypical antipsychotic
17 September 2015	Vraylar	Cariprazine	Schizophrenia and bipolar disorder	No	
10 July 2015	Rexulti	Brexpiprazole	Schizophrenia and major depressive disorder	No	
8 November 2013	Aptiom	Eslicarbazepine acetate	Seizures associated with epilepsy	No	
30 September 2013	Brintellix	Vortioxetine	Major depressive disorder	No	
22 October 2012	Fycompa	Perampanel	Partial onset seizures in patients with epilepsy	No	
10 June 2011	Potiga	Ezogabine	Seizures associated with epilepsy	No	
21 January 2011	Viibryd	Vilazodone hydrochloride	Major depressive disorder	No	
28 October 2010	Latuda	Lurasidone hydrochloride	Schizophrenia	No	
21 August 2009	Sabril	Vigabatrin	Complex partial seizures with or without secondary generalization	No	
13 August 2009	Saphris	Asenapine	Schizophrenia, acute manic or mixed episodes associated with bipolar	No	
6 May 2009	Fanapt	Iloperidone	Schizophrenia	No	
28 October 2008	Vimpat	Lacosamide	Partial onset seizure with epilepsy	No	
29 February 2008	Pristiq	Desvenlafaxine succinate	Major depressive disorder	Yes	Venlafaxine, an active metabolite of desvenlafaxine, was approved for depression treatment in 1993
19 December 2006	Invega	Paliperidone	Schizophrenia	Yes	Paliperidone is an active metabolite of the older antipsychotic risperidone
3 August 2004	Cymbalta	Duloxetine hydrochloride	Major depressive disorder	No	
26 November 2002	Strattera	Atomoxetine hydrochloride	Attention deficit hyperactivity disorder	No	
15 November 2002	Abilify	Aripiprazole	Schizophrenia	No	

**Table 2 ijms-24-11000-t002:** Development concepts and compound approaches of FDA-approved psychiatric drugs that were developed by drug repurposing between 2002 and 2022.

		Development Concept					
Compound Approach	Original Compound Type	Develops New Indication	Reduces Side Effect	Improves Symptom Control	Improves Dosing Compliance	Increases Efficacy	Reduces Drug–Drug Interaction
Use of original compound	Approved drug	Qelbree (2021) Lucemyra (2018)					
	Clinically developed compound	Diacomit (2018)					
	Component of natural products	Epidioloex (2018)					
Prodrug	Approved drug				Aristada (2015) Pristiq (2008)		Pristiq (2008)
Active metabolite	Approved drug		Invega (2006)				
Fixed-dose combination	Approved drugs					Auvelity (2022)	
	Approved and clinically developed compounds		Lybalvi (2021) KarXT (Ph3)				
	Approved drug and its prodrug			Azstarys (2021)			

( ): approved year except KarXT; development stage in KarXT.

## Data Availability

All data are available in the manuscript.

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
