# Peer review of "Advancements in Drug Repurposing: Examples in Psychiatric Medications"

_ijms, 2023, doi:10.3390/ijms241311000_

Round 1

Reviewer 1 Report

The communication entitled “Evolution of drug repurposing approaches and digital technologies in psychiatric drug development” describes the current update about the drug repurposing approaches for the psychiatric drug development. Drug repurposing aims not only to explore new applications, but also to reduce side effects, improve symptom control, improve drug compliance, enhance drug efficacy, and reduce drug-drug interactions. This sophisticated drug repurposing approaches, and a new assay system could accelerate the drug discovery. This article is very interesting. The study is comprehensive with solid literature evidence and manuscript is well written.

In summary, I recommend this manuscript for acceptance in International Journal of Molecular Sciences.

Reviewer 2 Report

I suggest to accept the manuscript in its current form.

As a very minor note, the SmartCube method is presented as somewhat magic approach. But, on the other hand, for the purpose of the review and taking into account sufficient citing the references for in depth study, it looks fine.

Reviewer 3 Report

The manuscript “Evolution of drug repurposing approaches and digital technologies in psychiatric drug development” examines developments over the past 20 years regarding drug repurposing strategies and new technologies that can estimate human efficacy from animal studies in psychiatric drug development. Drug repurposing is a hot-topic for the scientific community because it can help to find new treatments for diseases at a lower cost and in a shorter time, especially when preclinical safety studies have already been performed.
The article is well written and documented, but also has some weaknesses that could be improved in a revised version.
- It seems to me that there is a discrepancy between the title and the content of the article. The emphasis in the title is on the development of drugs for psychiatric use, and the content of the paper focus on drug repurposing methods.
- To better highlight the ramification and diversity of drug repurposing concepts, drug repurposing concept should be discussed in more detail in the introduction. I would recommend adding a section to define drug repurposing, introduce its main types, and provide some historical or current examples of success.
- The citation for SmartCube is missing.
- Table 1: I’m not sure if fospropofol and propofol are relevant examples for treating psychiatric diseases. Both are approved as short-term general anesthetics, and are used to induce and maintain general anesthesia. I am aware that there are studies suggesting that propofol may have rapid and lasting antidepressant effects in patients with treatment-resistant depression, but they are only preliminary and not sufficient to recommend propofol as a standard treatment for mental illness. I would suggest better explaining the reasoning behind their choice.
• Table 2: What side effects has the author consider wneh classified Aristada and Lusedr in "reduce side effects" category? For example, aripiprazole is an oral medication that can cause stomach problems such as nausea, vomiting, or indigestion. Aripiprazole lauroxil is an injectable drug that avoids these stomach problems, but it can cause problems at the injection site, such as pain, redness, or swelling. In addition, aripiprazole lauroxil stays in the body longer than oral aripiprazole, which means side effects may last longer. For example, an allergic reaction or movement disorder may be more difficult to treat with aripiprazole lauroxil than with oral aripiprazole. Similarly, fospropofol disodium is a water-soluble drug that does not cause pain when injected, but is slower and longer acting than propofol. It can also cause side effects such as tingling and itching, or more serious side effects that are harder to treat because propofol stays in the body longer.
-Line 202-205: “Another unique finding in this study is that for drugs approved before 2019, existing compounds were either tested as-is in separate applications or prodrug modification of existing compounds or utilization of active metabolites were used as drug repurposing approaches, whereas for all drugs approved in 2021 or later, fixed-dose combination approaches were used.”  
I would not make a rule of it. Viloxazine, was reapproved in 2021 for a completely different indication (conventional case of drug repurposing). Also, before 2018 there are many examples of drugs approved both as monotherapy and in combination with other active components (metformin vs metformin+sitagliptin, sitagliptin vs silagliptin+ertugliflozin, etc.). I would suggest rewording this statement so as not to mislead the reader.

Round 2

Reviewer 3 Report

  The author made satisfactory revisions to address my previous comments and I endorse this manuscript for publication in IJMS.